# Skin Carotenoid Level as an Alternative Marker of Serum Total Carotenoid Concentration and Vegetable Intake Correlates with Biomarkers of Circulatory Diseases and Metabolic Syndrome

**DOI:** 10.3390/nu12061825

**Published:** 2020-06-19

**Authors:** Mai Matsumoto, Hiroyuki Suganuma, Sunao Shimizu, Hiroki Hayashi, Kahori Sawada, Itoyo Tokuda, Kazushige Ihara, Shigeyuki Nakaji

**Affiliations:** 1Innovation Division, KAGOME CO., LTD. 17 Nishitomiyama, Nasushiobara 329-2762, Japan; Mai_Matsumoto@kagome.co.jp (M.M.); sunao_shimizu@kagome.co.jp (S.S.); Hiroki_Hayashi@kagome.co.jp (H.H.); 2Department of Vegetable Life Science, Hirosaki University Graduate School of Medicine, 5 Zaifu-cho, Hirosaki 036-8562, Japan; 3Center for Advanced Medical Science, Department of Stress Response Science, Hirosaki University Graduate School of Medicine, 5 Zaifu-cho, Hirosaki 036-8562, Japan; 4Department of Social Medicine, Hirosaki University Graduate School of Medicine, 5 Zaifu-cho, Hirosaki 036-8562, Japan; iwane@hirosaki-u.ac.jp (K.S.); i-tokuda@hirosaki-u.ac.jp (I.T.); ihara@hirosaki-u.ac.jp (K.I.); nakaji@hirosaki-u.ac.jp (S.N.)

**Keywords:** carotenoid, vegetable intake, circulatory disease, metabolic syndrome, non-invasive measurement, cross-sectional study

## Abstract

To confirm the usefulness of noninvasive measurements of skin carotenoids to indicate vegetable intake and to elucidate relationships between skin carotenoid levels and biomarkers of circulatory diseases and metabolic syndrome, we conducted a cross-sectional study on a resident-based health checkup (*n* = 811; 58% women; 49.5 ± 15.1 years). Skin and serum carotenoid levels were measured via reflectance spectroscopy and high-performance liquid chromatography, respectively. Vegetable intake was estimated using a dietary questionnaire. Levels of 9 biomarkers (body mass index [BMI], brachial-ankle pulse wave velocity [baPWV], systolic and diastolic blood pressure [SBP and DBP], homeostasis model assessment as an index of insulin resistance [HOMA-IR], blood insulin, fasting blood glucose [FBG], triglycerides [TGs], and high-density lipoprotein cholesterol [HDL-C]) were determined. Skin carotenoid levels were significantly positively correlated with serum total carotenoids and vegetable intake (*r* = 0.678 and 0.210, respectively). In women, higher skin carotenoid levels were significantly associated with lower BMI, SBP, DBP, HOMA-IR, blood insulin, and TGs levels and higher HDL-C levels. In men, it was also significantly correlated with BMI and blood insulin levels. In conclusion, dermal carotenoid level may indicate vegetable intake, and the higher level of dermal carotenoids are associated with a lower risk of circulatory diseases and metabolic syndrome.

## 1. Introduction

Do you know how many vegetables you ate yesterday or how many vegetables you consumed on average over the past month? Most consumers recognize the importance of consuming vegetables to maintain their health and prevent chronic diseases. However, few opportunities exist for consumers to calculate vegetable intake in their daily lives unless they participate in surveys on meal frequency. Thus, few consumers really know how many vegetables they eat.

Systematic reviews based on a meta-analysis of prospective cohort studies have shown that sufficient dietary intake of vegetables and fruits can effectively reduce mortality from all causes [1] and prevent ischemic heart disease and cerebrovascular disease [2,3,4]. The World Health Organization recommends aggressive vegetable and fruit intake by consuming more than 400 g of fruits and vegetables daily to improve overall health and reduce the risk of certain noncommunicable diseases, including cardiovascular diseases and some cancers [5]. Many countries also recommend this and set goals for fruit and vegetable consumption [6,7,8,9]. However, actual vegetable intake does not meet these goals in many countries [6]. For example, the Ministry of Health, Labour and Welfare of Japan recommends that people consume 350 g of vegetables daily [10], but the average vegetable intake among Japanese people is only around 280–290 g and has not changed over the past two decades [11]. Presenting vegetable and fruit intake in a simple quantitative manner and communicating the relationship between vegetable and fruit intake and health will make these recommendations more meaningful to consumers.

An ingredient specific to vegetables and fruits that is accumulated but not biosynthesized in the human body may be a potential indicator for quantifying vegetable and fruit intake. Noninvasive measurements of carotenoids [12,13], which are abundant in vegetables (e.g., tomatoes, carrots, and spinach) and fruits (e.g., citrus and prunes), may be a potential candidate. These values can be measured via resonance Raman spectroscopy (RRS) [14] and have been shown to correlate with fruit and vegetable intake [15,16,17,18,19]. However, RRS instruments are too expensive to popularize this method. In recent years, techniques using a reflective spectrophotometry (RS) have been developed and are highly correlated with the RRS method [20,21]. Intake of carotenoid-rich beverages, supplements, and foods increases the dermal carotenoid levels measured via RS [22,23,24]. Thus, RS may enable estimating vegetable and fruit intake.

Many studies have revealed significant relationships between serum carotenoid concentrations and biomarkers of chronic circulatory diseases and metabolic syndrome [25,26,27,28]. Obana et al. suggested that dermal carotenoids were negatively correlated with hemoglobin and positively correlated with blood albumin and total cholesterol [29]; however, few reports have suggested an association between dermal carotenoid levels and health status. Therefore, we noninvasively evaluated skin carotenoid levels measured during an annual health checkup in Japan (i.e., the Iwaki Health Promotion Project 2018) for (1) correlation with serum carotenoids, (2) correlation with vegetable and citrus fruit intake, and (3) association with biomarkers of chronic circulatory diseases and metabolic syndrome.

## 2. Materials and Methods

### 2.1. Study Design and Subjects

This population-based cross-sectional study was conducted during an annual health checkup, the 2018 Iwaki Health Promotion Project in Japan, from 26 May to 4 June 2018. Of the 1056 participants aged ≥20 years, 811 (340 men and 471 women) were recruited as relatively healthy subjects. These participants had complete clinical data, were not taking medication for dyslipidemia, and had no history of serious diseases such as cancer, stroke, cardiovascular diseases, kidney diseases, liver diseases, or diabetes. The study procedures and subject recruitment were conducted in accordance with the Declaration of Helsinki and were approved by the ethics boards of Hirosaki University School of Medicine (2018-012, 2018-063) and KAGOME CO., LTD (2018-R02). All subjects provided written informed consent.

### 2.2. Self-Administered Questionnaire

Self-administered questionnaires were issued to the participants in advance of the health checkup and collected on the day of blood collection. The questionnaire obtained information regarding sex, age, current smoking status, current exercise habits, medical history, and medication use. Subjects who engaged in exercise for ≥30 min per day at least twice weekly habitually throughout the year were defined as being accustomed to exercise. The frequency of food and drink consumption was surveyed using the Brief-type self-administered diet history questionnaire (BDHQ) [30], and the daily alcohol, vegetable and citrus fruit intake were calculated from this. Total vegetable intake was expressed by the sum of green and yellow vegetables (i.e., green vegetables, carrots and pumpkins, tomatoes, and pickled green vegetables) and light-colored vegetables (i.e., daikon radish and kabu, cabbage, root vegetables, and pickles (except those of green leafy vegetables)).

### 2.3. Body Measurement

Body mass index (BMI) was calculated from body weight and height in light clothing without shoes. Brachial-ankle pulse wave velocity (baPWV), an index of arteriosclerosis, was measured by a volume-plethysmograph (Form PWV/ABI, OMRONCOLIN Co Ltd., Tokyo, Japan), and the average of the right and left sides was used for the analysis. Systolic blood pressure (SBP) and diastolic blood pressure (DBP) were measured using a mercury sphygmomanometer with participants sitting in chairs.

### 2.4. Blood Sampling and Testing

Blood was collected from the cubital median vein after overnight fasting. Measuring of diabetes- and dyslipidemia-related blood biomarkers (i.e., blood insulin, fasting blood glucose (FBG), triglycerides (TGs), and high-density lipoprotein cholesterol (HDL-C)) was commissioned to LSI Medience Co., Ltd. (Tokyo, Japan). FBG, TGs, and HDL-C concentrations were measured using enzyme assays. Blood insulin concentrations were determined via chemiluminescence immunoassays. The homeostasis model assessment of insulin resistance (HOMA-IR) values were calculated from the blood insulin concentrations and FBG. Blood antioxidants were measured by KAGOME CO., LTD. Serum carotenoids (lutein, zeaxanthin, β-cryptoxanthin, α-carotene, β-carotene, and lycopene), α-tocopherol (vitamin E), and retinol (vitamin A) were extracted from serum as per the method of Oshima et al. [31] and quantified via simultaneous assays using a high-performance liquid chromatograph (HPLC) with a photodiode array detector (Prominence LC-30AD/Nexera X2 SPPD-M30A, Shimadzu Co., Kyoto, Japan) [32]. Plasma ascorbic acid (vitamin C) concentrations were determined using a commercially available kit (R01K02, SHIMA Laboratory Co., Ltd., Tokyo, Japan) based on a colorimetric method.

### 2.5. Measurement of Skin Carotenoids

Skin carotenoid levels were measured using a Multiple Spatially Resolved Reflection Spectroscopy Sensor (Biozoom Services GmbH, Kassel, Germany) based on the RS method [22]. Light at 350–1000 nm was provided via 118 LED light emitters in 16 steps, and the sensor detected light reflected by the skin at 152 light-sensitive areas. Algorithms representing the skin carotenoid levels (0.1~12.0) were designed to ensure optimal correlation with the measured RRS values. The skin was measured (once per person) on the palm side of the thumb base, completely covering the sensor, without allowing stray light to enter. In case any error message was displayed, it was remeasured.

### 2.6. Statistical Analysis

The Mann–Whitney U test was used to compare the median value of each measurement between men and women; Fisher’s exact test was used to compare categorical variables, including current smoking habits, exercise habits, and antihypertensive medications. Simple correlation analysis between skin carotenoid levels and serum carotenoid levels or between these levels and intake of total, green-yellow, and light-colored vegetables was performed using Pearson’s correlation coefficient. Z-transformation was used to compare correlation coefficients. Multiple linear regression analyses were also performed to evaluate the contributions of skin and serum carotenoid levels and sex in vegetable intake. The relationships between skin carotenoid levels or serum total carotenoid concentrations and biomarkers of circulatory disease or metabolic syndrome were analyzed by multiple regression analysis using the former as an explanatory variable and the latter as a response variable. Age, alcohol consumption, current smoking status, current exercise habits, and use of antihypertensive medications were used as explanatory variables for adjustment. Vitamins (i.e., retinol, ascorbic acid, and α-tocopherol) were added to the regulators when total carotenoid concentrations were included as explanatory variables. Because blood carotenoid concentrations and blood vitamins were not normally distributed, logarithmic values were used for the analyses. In addition, for each model obtained from the multiple regression analysis, Cook’s distance was calculated, and samples with Cook’s distances ≥0.5 were excluded for reanalysis. A significance level of *p* < 0.05 was adopted for all analyses, which were performed using R statistical software (version 3.5.0, R Foundation for Statistical Computing, Vienna, Austria).

## 3. Results

### 3.1. Characteristics of Subjects

Table 1 presents the mean values of the parameters measured for both the subjects as a whole and stratified by sex. For lifestyles and antihypertensive medication rates, smoking rate and alcohol intake were significantly higher in men, but exercise habit percentages and antihypertensive medication compliance rates did not differ between sexes. Skin carotenoid levels and serum concentrations of five of the six major carotenoids (excluding zeaxanthin) and total serum carotenoids were significantly higher in women. Daily intakes of total vegetables and green and yellow vegetables were also higher in women. Of the nine biomarkers of circulatory diseases and metabolic syndrome, HOMA-IR and blood insulin levels did not differ by sex, but averages of the other seven biomarkers differed between sexes; HDL-cholesterol was significantly lower, and the other 6 biomarkers were higher in men.

### 3.2. Correlation between Skin Carotenoid Levels and Serum Carotenoid Concentrations

Skin carotenoid levels were significantly positively correlated with total serum carotenoid concentrations and all six individual carotenoids for all subjects and for subjects stratified by sex (Table 2). Total serum carotenoid concentrations had the highest correlation coefficient to skin carotenoid levels for all subjects and both sexes, whereas individual carotenoids did not. For individual carotenoids, the r values of the correlations with β-carotene and α-carotene were relatively high. The r value for lycopene alone was higher in men; all other r values were higher in women.

### 3.3. Correlation between Vegetable Intake and Skin Carotenoid Levels or Total Serum Carotenoid Concentrations

Skin carotenoid levels, as well as serum total carotenoid concentrations, were significantly positively correlated with total vegetables, green and yellow vegetables, light-colored vegetables, and fruit intake as calculated from the BDHQ (Table 3). The correlation coefficient for total vegetable intake in the total subjects was low (r = 0.210) but was not significantly lower (Z = 1.651, *p* = 0.099) than that between total serum carotenoids and total vegetable intake (r = 0.287). Skin carotenoids were significantly positively correlated with total vegetable, green and yellow vegetable, and light-colored vegetable intake but not with fruit intake, when stratified by sex. Multiple linear regression analyses indicated that skin carotenoid levels and serum total carotenoid concentrations were significantly associated with total, green and yellow, and light-colored vegetables, and the effect of sex was not significant (Table 4). No significant interaction was observed between skin and serum carotenoid levels and sex.

### 3.4. Correlation with Markers of Circulatory Diseases and Metabolic Syndrome

Table 5 shows the relationship between skin carotenoid levels and serum total carotenoid levels and biomarkers associated with circulatory diseases and metabolic syndrome. Multiple regression analyses adjusted for age, use of antihypertensive medication, smoking habits, alcohol intake, exercise habits, and blood vitamin A (retinol), C (ascorbic acid), and E (α-tocopherol) levels showed that skin carotenoid levels were significantly positively correlated with HDL-C and significantly negatively correlated with BMI, SBP, DBP, HOMA-IR, blood insulin, and TGs in women. The negative correlation with SBP and DBP was not observed for serum total cholesterol. In men, skin carotenoid levels were significantly correlated with only BMI and blood insulin. In contrast, serum total carotenoid concentration was correlated with BMI, HOMA-IR, blood insulin, TGs, and HDL-C.

## 4. Discussion

Skin carotenoid levels are highly correlated with serum carotenoid concentrations and are potential indicators of vegetable intake, which is encouraged to help prevent circulatory diseases. However, no study has simultaneously examined the relationships between skin carotenoid levels and serum carotenoid concentrations, vegetable intake, and predictive biomarkers of chronic diseases. To our knowledge, this study is the first to demonstrate that skin carotenoid levels reflect vegetable intake and that the levels were significantly associated with multiple predictive biomarkers of circulatory diseases and metabolic syndrome in relatively healthy adults.

Skin carotenoid levels measured in this study were positively correlated with serum total carotenoid concentrations (Table 2). The correlation coefficient was high (0.678) and was comparable to that reported by Jahns et al. (0.70) [16] and Meinke et al. (0.6672) [19] although the measurement of our study was once per person and it contained some uncertainty. This suggests that skin carotenoid levels may be noninvasive measurable indicators of total serum carotenoid concentrations. Skin carotenoid levels were significantly positively correlated with both total serum carotenoids and individual serum carotenoid concentrations. However, the correlation coefficient for each carotenoid was lower than that for total carotenoids. The measurements used in this study were designed with an algorithm that optimizes the correlation with RRS measurements [20,21], and RRS detects conjugated double-bond chains common to carotenoids [33]. Therefore, the correlation coefficient between skin carotenoid levels and total serum carotenoid concentrations might be the highest. For each individual serum carotenoid, the correlation coefficient with skin carotenoid levels differed. β-carotene, which was the most common carotenoid in the participants’ serum and is widely present in many vegetables, yielded the highest correlation coefficients, and zeaxanthin, the least common carotenoid in the serum, had the lowest correlation coefficients. Lycopene was the third most abundant serum carotenoid among participants, but the correlation coefficient was relatively low (r = 0.278). The Raman intensities of lycopene and β-carotene, as measured by RRS, were nearly the same under argon laser excitation at 488 nm [33]. However, the maximum absorption wavelength of lycopene differed from the other 5 carotenoids [34] and thus might affect the RS measurement levels and consequently might correlate weakly for lycopene. Correlations have been reported between plasma carotenoid concentrations and carotenoid concentrations in biopsied skin [35,36], but the composition ratio of each carotenoid in the plasma and skin differ. Capsanthin, a xanthophyll from plasma, such as lutein and β-cryptoxanthin, was reported to clear much faster than lycopene [31]. Meinke et al. suggested that increased skin carotenoid levels by continuous intake of vegetable extract was observed posterior to that in the blood [22]. These differences in accumulation and clearance of carotenoids by various tissues could have caused the different correlation coefficients for each carotenoid in this study. In general, carotenoids show strong quenching activity for singlet oxygen, a reactive oxygen species. Measuring internal carotenoid levels may help explain health conditions, and skin carotenoid levels could be good indicators.

The correlation between skin carotenoid levels and total vegetable intake, as determined by the BDHQ, was weak at 0.210 (Table 3). However, it did not significantly differ from that between serum total carotenoid concentrations and total vegetable intake (0.287). Thus, as an alternative indicator of vegetable intake, skin carotenoid levels are not significantly inferior to serum total carotenoid concentrations. We speculate that we had a low correlation coefficient owing to limitations due to the uncertainty inherent in the dietary frequency method of the BDHQ [30], although this method has been validated. Jahns et al. [19] reported that the correlation coefficients between skin carotenoid levels measured via RS and 24-h recall vegetable and fruit intake were 0.27 (not significant) and 0.37 (*p* < 0.01), respectively, for baseline across the year, showing similar values to the coefficient of the present results. As green and yellow vegetables contain abundant carotenoids, the correlation coefficient between them was the highest in the present study. We assessed the relationship between skin carotenoid levels and vegetable intake, whereas most previous RS studies assessed the correlation between vegetables and fruits. Citrus fruits contain many carotenoids, but the correlation coefficient was lower than that of light-colored vegetables poor in carotenoids, and the correlation was not significant when subjects were stratified by sex. The most popular citrus fruit in Japan is the Satsuma mandarin, and the survey was conducted out of season; thus, the subjects’ citrus intake was much less than their vegetable intake.

Relationships between serum carotenoid concentrations and biomarkers of circulatory diseases and metabolic syndrome have been reported in cross-sectional studies [25,26,37] as well as prospective cohort studies [27,38]. We also confirmed that serum carotenoid concentrations significantly correlated with many of the biomarkers measured in this study on the same resident-based health check-up during 2015 to 2018 [39]. Because skin carotenoid levels are highly positively correlated with serum total carotenoids, they may show similar relationships to biological markers, but, to our knowledge, no study has assessed the relationships with multiple biomarkers except a study conducted by Obana et al. [29]. The associations observed in our study (Table 5) might suggest that higher skin carotenoid levels are associated with a lower risk of chronic circulatory diseases and metabolic syndrome. However, the skin carotenoid index measured by Obana et al. was significantly positively correlated with total cholesterol and blood albumin and negatively correlated with hemoglobin; it was not correlated with TGs, HDL-C, low-density lipoprotein (LDL)-C, or HbA1c. Most subjects in that study were patients at an ophthalmology clinic and were older than our subjects (mean: 69.7 ± 13.6 vs. 49.5 ± 15.1 years, respectively). Our study was conducted on relatively healthy subjects, which may better enable assessing the associations between skin carotenoid levels and biomarkers of circulatory diseases and metabolic syndrome.

The number of biomarkers that were significantly correlated with skin carotenoid levels varied greatly between men and women in the present study, with only two of the nine biomarkers being significant in men, compared with seven in women. The mean skin carotenoid levels were significantly higher in women than in men (Table 1), and many studies have shown high carotenoid levels in the blood and skin in women [29,40,41], although the mechanism is unclear. Higher bioavailability of carotenoids in women might be one reason that more significant correlations were observed in women. Skin carotenoid levels were significantly negatively correlated with SBP and DBP in women, although serum total carotenoids were not. Among the carotenoids evaluated in this study, a meta-analysis suggested that lycopene intake significantly reduced SBP [42,43]. However, the detection sensitivity of lycopene is relatively low via RS. Some ingredients in vegetables in addition to carotenoids, such as potassium, γ-aminobutyric acid, nicotianamine, and other ingredients, could help lower blood pressure [44,45,46], although the correlation coefficient between skin carotenoid levels and vegetable intake was not higher than that between serum total carotenoids and vegetable intake. Further studies are required to reveal the cause of the significant association.

Inhibition of carbohydrate absorption by vegetable dietary fiber [47,48,49] is assumed to have caused the negative correlation with blood insulin in both men and women. Additionally, oxidative stress causes abnormal insulin secretion in pancreatic β-cells, and the antioxidant effect of carotenoids may have protected the pancreatic β-cells [50,51]. The positive correlation with HDL-C in women is presumed to be due to the distribution of carotenoids in the blood. Carotenoids are transported primarily in HDL-C and LDL-C particles when transported in the blood. Xanthophylls, such as lutein and β-cryptoxanthin, are highly partitioned into HDL-C particles [31,52], and the role of HDL-C particles as xanthophyll pools may be a factor. Furthermore, lycopene is reported to increase HDL-2 and HDL-3 by increasing lecithin cholesterol acyltransferase activity [53], and lycopene and β-carotene decrease LDL-C and indirectly increase HDL-C by inhibiting HMG-CoA reductase, the rate-limiting enzyme in cholesterol synthesis [54,55]. Further investigations are required to elucidate the contribution and potential causality of these mechanisms.

## 5. Conclusions

Skin carotenoid levels were highly correlated with serum carotenoid concentrations and thus can be used as an alternative indicator for estimating vegetable intake. We showed for the first time that skin carotenoid levels were significantly associated with biomarkers of circulatory diseases and metabolic syndrome in the general population without apparent illness. Thus, noninvasively measuring skin carotenoid levels quantifies vegetable intake and can enable determining possible vegetable deficiencies during health checkups. The causal relationship among biological markers remains to be examined, but these results provide a reference for the risk of chronic diseases in healthy adults.

## Figures and Tables

**Table 1 nutrients-12-01825-t001:** Comparison of measurements by sex.

	Total	Male	Female	
Number of Subjects	811	340	471	
Age (years)	49.5 ± 15.1	48.8 ± 14.6	50.1 ± 15.4	
Current Smoking (%)	19 ± 0.39	31 ± 0.46	10 ± 0.30	***
Habitual Exercise (%)	8 ± 0.27	9 ± 0.28	7 ± 0.26	
Alcohol Intake (mL/d)	13.75 ± 22.86	24.90 ± 28.00	5.64 ± 13.30	***
Antihypertensive (%)	18 ± 0.38	20 ± 0.40	17 ± 0.37	
Skin Carotenoid	5.41 ± 1.30	4.78 ± 1.08	5.87 ± 1.25	***
Serum Carotenoid				
Total Carotenoid (µg/mL)	1.286 ± 0.653	1.060 ± 0.549	1.460 ± 0.662	***
Lutein (µg/mL)	0.304 ± 0.146	0.282 ± 0.128	0.319 ± 0.156	***
Zeaxanthin (µg/mL)	0.064 ± 0.026	0.066 ± 0.025	0.063 ± 0.027	
β-Cryptoxanthin (µg/mL)	0.123 ± 0.086	0.094 ± 0.065	0.144 ± 0.093	***
α-Carotene (µg/mL)	0.141 ± 0.142	0.113 ± 0.146	0.161 ± 0.136	***
β-Carotene (µg/mL)	0.414 ± 0.333	0.271 ± 0.253	0.517 ± 0.346	***
Lycopene (µg/mL)	0.250 ± 0.151	0.236 ± 0.152	0.260 ± 0.149	**
Vegetable and fruit Intake				
Total vegetable (g/day)	184.2 ± 116.3	170.0 ± 109.0	194.0 ± 120.0	**
Green vegetables (g/day)	74.1 ± 56.1	66.1 ± 51.6	79.9 ± 58.5	***
Light-colored vege. (g/day)	110.1 ± 70.0	104.0 ± 65.3	114.0 ± 73.0	
Citrus (g/day)	9.0 ± 16.2	7.7 ± 14.5	10.0 ± 17.3	
BMI (kg/m^2^)	22.7 ± 3.5	23.9 ± 3.4	21.9 ± 3.4	***
baPWV (cm/sec)	1370 ± 316	1420 ± 304	1330 ± 320	***
SBP (mmHg)	123.2 ±18.2	127.0 ± 18.0	121.0 ± 17.9	***
DBP (mmHg)	78.2 ± 11.9	81.5 ± 12.2	75.8 ± 11.1	***
HOMA-IR	1.25 ± 0.88	1.34 ± 1.15	1.18 ± 0.61	
Insulin (µU/mL)	5.32 ± 2.87	5.52 ± 3.49	5.18 ± 2.31	
FBG (mg/dL)	92.4 ± 12.9	95.1 ± 15.6	90.5 ± 10.1	***
TGs (mg/dL)	93.6 ± 68.7	118.0 ± 90.0	76.1 ± 39.2	***
HDL (mg/dL)	65.7 ± 17.7	59.1 ± 16.6	70.4 ± 17.0	***

Green vegetables: green and yellow vegetables, Light-colored vege.: light-colored vegetables, BMI: body mass index, baPWV: brachial-ankle pulse wave velocity, SBP: systolic blood pressure, DBP: diastolic blood pressure, HOMA-IR: homeostasis model assessment as an index of insulin resistance, Insulin: blood insulin, FBG: fasting blood glucose, TGs: triglycerides, HDL: high-density lipoprotein cholesterol. Fisher’s exact test was used to compare current smoking, exercise habit, and antihypertensive medication rates by sex. The Mann–Whitney U test was applied for all other measurements. ** *p* < 0.01, *** *p* < 0.001.

**Table 2 nutrients-12-01825-t002:** Correlation between skin carotenoid levels and serum carotenoid concentrations.

	Total (*n* = 811)		Male (*n* = 340)		Female (*n* = 471)	
	r	*p*	r	*p*	r	*p*
Total Carotenoid	0.678	<0.001	0.587	<0.001	0.653	<0.001
Lutein	0.414	<0.001	0.348	<0.001	0.431	<0.001
Zeaxanthin	0.164	<0.001	0.184	<0.001	0.220	<0.001
β-Cryptoxanthin	0.475	<0.001	0.339	<0.001	0.417	<0.001
α-Carotene	0.615	<0.001	0.545	<0.001	0.580	<0.001
β-Carotene	0.653	<0.001	0.516	<0.001	0.626	<0.001
Lycopene	0.278	<0.001	0.297	<0.001	0.234	<0.001

Pearson’s correlation coefficient (r) with *p* values are shown.

**Table 3 nutrients-12-01825-t003:** Correlation between vegetable intake and (1) skin carotenoid levels and (2) serum total carotenoid concentrations.

	Total (*n* = 811)		Male (*n* = 340)		Female (*n* = 471)	
	*r*	*p*	*r*	*p*	*r*	*p*
(1) Skin Carotenoid						
Total Vegetable	0.210	<0.001	0.169	<0.01	0.194	<0.001
Green vegetables	0.217	<0.001	0.175	<0.01	0.188	<0.001
Light-colored vege.	0.174	<0.001	0.144	<0.01	0.169	<0.001
Citrus	0.113	<0.01	0.096	NS	0.090	NS
(2) Serum Total Carotenoid						
Total Vegetable	0.287	<0.001	0.262	<0.001	0.275	<0.001
Green vegetables	0.350	<0.001	0.266	<0.001	0.293	<0.001
Light-colored vege.	0.232	<0.001	0.227	<0.001	0.219	<0.001
Citrus	0.173	<0.001	0.228	<0.001	0.117	<0.05

Pearson’s correlation coefficient (r) and the *p* values are shown. Green vegetables: green and yellow vegetables, Light-colored vege.: Light-colored vegetables, NS: not significant.

**Table 4 nutrients-12-01825-t004:** Association between vegetable intake and (1) skin carotenoid levels and (2) total serum carotenoid concentrations (Multiple regression analysis).

	Carotenoid Level		Gender		Interaction	
	std. β	*p*	std. β	*p*	std. β	*p*
(1) Skin Carotenoid						
Total Vegetable	0.193	<0.01	0.035	NS	0.016	NS
Green vegetables	0.195	<0.01	0.078	NS	0.010	NS
Light-colored vege.	0.164	<0.05	−0.005	NS	0.019	NS
Citrus	0.104	NS	0.057	NS	−0.004	NS
(2) Serum Total Carotenoid						
Total Vegetable	0.256	<0.001	0.004	NS	0.058	NS
Green vegetables	0.254	<0.001	0.037	NS	0.082	NS
Light-colored vege.	0.221	<0.001	−0.023	NS	0.030	NS
Citrus	0.213	<0.001	0.008	NS	−0.076	NS

Multiple regression analysis results are represented as standardized partial regression coefficients (std. β) and *p* values. Green vegetables: green and yellow vegetables, Light-colored vege.: light-colored vegetables, NS: not significant.

**Table 5 nutrients-12-01825-t005:** Associations between skin and serum total carotenoids and biomarkers of circulatory diseases and metabolic syndrome.

	Skin Carotenoid				Serum Total Carotenoid			
	Male		Female		Male		Female	
	std. β	*p*	std. β	*p*	std. β	*p*	std. β	*p*
BMI	−0.120	<0.05	−0.160	<0.001	−0.219	<0.001	−0.262	<0.001
baPWV	−0.023	NS	−0.056	NS	−0.060	NS	−0.089	<0.05
SBP	−0.032	NS	−0.123	<0.01	−0.093	NS	−0.057	NS
DBP	−0.058	NS	−0.095	<0.05	−0.097	NS	−0.055	NS
HOMA-IR	−0.108	NS	−0.122	<0.05	−0.202	<0.01	−0.236	<0.001
Insulin	−0.127	<0.05	−0.125	<0.01	−0.236	<0.001	−0.243	<0.001
FBG	−0.019	NS	−0.015	NS	−0.085	NS	−0.064	NS
TGs	−0.042	NS	−0.141	<0.01	−0.284	<0.001	−0.251	<0.001
HDL	−0.047	NS	0.134	<0.01	0.183	<0.01	0.337	<0.001

Associations between skin carotenoid levels and serum total carotenoid concentrations and circulatory diseases and metabolic syndrome via multiple regression analysis are represented as standardized partial regression coefficients and *p* values. BMI: body mass index, baPWV: brachial-ankle pulse wave velocity, SBP: systolic blood pressure, DBP: diastolic blood pressure, HOMA-IR: homeostasis model assessment as an index of insulin resistance, Insulin: blood insulin, FBG: fasting blood glucose, TGs: triglycerides, HDL-C: high-density lipoprotein cholesterol, std β: standardized partial regression coefficient, NS: not significant.

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
