# Peer review of "Skin Carotenoid Level as an Alternative Marker of Serum Total Carotenoid Concentration and Vegetable Intake Correlates with Biomarkers of Circulatory Diseases and Metabolic Syndrome"

_nutrients, 2020, doi:10.3390/nu12061825_

Round 1
Reviewer 1 Report
This is a very well carried study that was part of a larger health study in Japan. I only have two minor comments that I have addressed to the authors.
1) On line 120 measurement of skin carotenoids.
Please provide a range of reading for the RS instrument that I believe it only provides a number. Also, how many measurements were carried out on each individual and what was the variability of measurements for each individual. The RS instrument sometimes provides different reading for the same subject after a second or a third measurement.
2) On line 169 it is mentioned that individual carotenoids measured in skin by RS instrument do not correlate with serum levels. I don't believe that the RS instrument can distinguish between different skin carotenoids and measure each of these since in most cases the absorption maximum of carotenoids is at 446-454 nm with the exception of lycopene that is at 470-474 nm.
Author Response
Responses to the comments of Reviewer #1
(Revisions in the text are shown using yellow highlight and red ink for additions, and strikethrough font [example] for deletions.)
- On line 120 measurement of skin carotenoids. Please provide a range of reading for the RS instrument that I believe it only provides a number. Also, how many measurements were carried out on each individual and what was the variability of measurements for each individual. The RS instrument sometimes provides different reading for the same subject after a second or a third measurement.
Response: Thank you very much for providing important insights. According to your suggestion, we added the range of measurements and the number of times of the measurement per person in the “Materials and Methods” as follows;
Line 124: Algorithms representing the skin carotenoid levels (0.1 ~ 12.0) were designed to ensure optimal correlation with the measured RRS values. The skin was measured (once per person) on the palm side of the thumb base, completely covering the sensor, without allowing stray light to enter. In case any error message was displayed, it was remeasured.
We additionally mentioned about the uncertainty derived from that we only measured one time per person in the “Discussion”.
Line 226: The correlation coefficient was high (0.678) and was comparable to that reported by Jahns et al. (0.70) [16] and Meinke et al. (0.6672) [19] although the measurement of our study was once per person and it contained some uncertainty.
- On line 169 it is mentioned that individual carotenoids measured in skin by RS instrument do not correlate with serum levels. I don't believe that the RS instrument can distinguish between different skin carotenoids and measure each of these since in most cases the absorption maximum of carotenoids is at 446-454 nm with the exception of lycopene that is at 470-474 nm.
Response: We appreciate your comment on this point. Our result suggested that skin carotenoid levels (range: 0.1 ~ 12.0) significantly positively correlated with not only serum concentration of total carotenoid but also that of each carotenoid. But, the correlate coefficient (r) of each serum carotenoid was lower than that of total carotenoid (Table 2). We looked at this point in the “Discussion” on the Line 231~.
We recognized that RS instrument could NOT distinguish each carotenoid as you mentioned. The difference in correlate coefficients between skin carotenoid level and each serum carotenoid may highly depend on its abundance ratio which is affected by what we eat. We think the results of correlation between skin carotenoid level and serum concentration of each carotenoid should be meaningful to understand the difference of carotenoid bioavailability and what we should eat to keep skin carotenoid level high.
Thank you again for your comments on our paper. I trust that the revised manuscript is suitable for publication.
Reviewer 2 Report
Drs. Matsumoto, Suganuma et al. present here the compelling use of skin carotenoid levels--as measured by reflectance spectroscopy--as a surrogate for serum carotenoid levels. This technology makes objective carotenoid levels more accessible to clinicians.
The manuscript is incredibly interesting and well-written. The background is well-developed, the methods are sound, the results are robust and sound, and the conclusions are significant and supported by their methods and results.
I do have some style/English edits that should be considered:
MAJOR EDITS:
- Pg. 2, Lines 56-57: I am unsure of the meaning of this last part. Do the authors mean "...will make these recommendations more meaningful to consumers."? Please review meaning and edit for clarity.
- Pg. 5, Line 176 (Table 2): Please change "R" to "r" and "P" to "p" for consistency.
MINOR EDITS:
- Pg. 1, Lines 25-29: Suggest changing "( )" inside the parentheses to "[ ]". That is more standard as done in other portions of the manuscript.
- Pg. 2, Line 55: Please change "2" to "two". If less than "10", write out the number.
- Pg. 2, Line 81: I suggest "recruited" instead of "extracted"
- Pg, 4, Table 1: I suggest spelling out "Green vegetables (g/day)". It will fit in the space in the table and makes the table easier to read.
Thank you for the opportunity to review this manuscript.
Author Response
Responses to the comments of Reviewer #2
(Revisions in the text are shown using yellow highlight and red ink for additions, and strikethrough font [example] for deletions.)
I do have some style/English edits that should be considered:
MAJOR EDITS:
- Pg. 2, Lines 56-57: I am unsure of the meaning of this last part. Do the authors mean "...will make these recommendations more meaningful to consumers."? Please review meaning and edit for clarity.
- Pg. 5, Line 176 (Table 2): Please change "R" to "r" and "P" to "p" for consistency
MINOR EDITS:
- Pg. 1, Lines 25-29: Suggest changing "( )" inside the parentheses to "[ ]". That is more standard as done in other portions of the manuscript.
- Pg. 2, Line 55: Please change "2" to "two". If less than "10", write out the number.
- Pg. 2, Line 81: I suggest "recruited" instead of "extracted"
- Pg, 4, Table 1: I suggest spelling out "Green vegetables (g/day)". It will fit in the space in the table and makes the table easier to read.
Response: We wish to thank the reviewer for these comments to improve English of our manuscript. In accordance with the suggestion, we revised our manuscript as follows;
MAJOR EDIT #1: The reviewer’s understanding was what we’d like to mention.
Pg. 2, Lines 56-57: Presenting vegetable and fruit intake in a simple quantitative manner and communicating the relationship between vegetable and fruit intake and health will allow more easily making make these recommendations more meaningful to consumers.
MAJOR EDIT #2: We changed Table 2 according to the reviewer’s suggestion.
Pg. 5, Table 2: “R” to “r”, and “P” to “p”.
MINOR EDITS #1 - #3: We changed our manuscript according to the reviewer’s suggestion.
Pg. 1, Lines 25-29: Levels of 9 biomarkers (body mass index ([BMI)], brachial-ankle pulse wave velocity ([baPWV)], systolic and diastolic blood pressure ([SBP and DBP)], homeostasis model assessment as an index of insulin resistance ([HOMA-IR)], blood insulin, fasting blood glucose ([FBG)], triglycerides ([TGs)], and high-density lipoprotein cholesterol ([HDL-C)]) were determined.
Pg. 2, Line 55: past 2 two decades
Pg. 2, Line 81: were extracted recruited
MINOR EDIT”4: Based on the reviewer’s suggestion, we changed from “Green vege.” to “Green vegetables” not only in the Table 1, but also Table 3 and the footnotes.
Thank you again for your comments on our paper. I trust that the revised manuscript is suitable for publication.